# Barriers to and Facilitators for Accessing HPV Vaccination in Migrant and Refugee Populations: A Systematic Review

**DOI:** 10.3390/vaccines12030256

**Published:** 2024-02-29

**Authors:** Davide Graci, Nicolò Piazza, Salvatore Ardagna, Alessandra Casuccio, Anton Drobov, Federica Geraci, Angelo Immordino, Alessandra Pirrello, Vincenzo Restivo, Riccardo Rumbo, Rosalba Stefano, Roberta Virone, Elena Zarcone, Palmira Immordino

**Affiliations:** 1Department of Health Promotion, Mother and Childcare, Internal Medicine and Medical Specialties “G. D’Alessandro”, University of Palermo, Via del Vespro 133, 90127 Palermo, Italy; davide.graci01@unipa.it (D.G.); nicolo.piazza@unipa.it (N.P.); alessandra.casuccio@unipa.it (A.C.); federica.geraci01@unipa.it (F.G.); riccardo.rumbo@unipa.it (R.R.); rosalba.stefano@you.unipa.it (R.S.); roberta.virone@unipa.it (R.V.); elena.zarcone@unipa.it (E.Z.); 2Department of Public Health, Faculty of Medicine, Masaryk University, 625 00 Brno, Czech Republic; anton.drobov@med.muni.cz; 3Department of Biomedicine, Neuroscience and Advanced Diagnostics, AOUP Paolo Giaccone, University of Palermo, Via del Vespro 133, 90127 Palermo, Italy; angelo.immordino182@gmail.com; 4School of Medicine, University Kore of Enna, 94100 Enna, Italy; vincenzo.restivo@unikore.it; 5Centro di Ateneo “MIGRARE”, Università degli Studi di Palermo, 90133 Palermo, Italy

**Keywords:** HPV, prevention, vaccination, vaccine hesitancy, social determinants, vaccination strategies, public health, migrants, refugees

## Abstract

Human papillomavirus (HPV) is the most prevalent sexually transmitted virus globally and a primary cause of cervical cancer, which ranks fourth among tumors in both incidence and mortality. Despite the availability of effective vaccines worldwide, HPV vaccination rates vary, especially among migrant and refugee populations. Indeed, migrant status may act as a determinant against accessing vaccinations, among many other factors. The objective of this paper is to evaluate barriers to and facilitators for accessing HPV vaccination in migrant and refugee populations. A systematic review of the existing peer-reviewed academic literature was conducted according to the PRISMA 2020 guidelines in which we examined thirty-four studies to evaluate HPV vaccination rates in these populations and identify factors acting as barriers or facilitators. Key determinants include socio-economic status and health literacy. Communication barriers, including language and cultural factors, also impact access to information and trust in the health workforce. Understanding and considering these factors is crucial for developing proper and inclusive vaccination strategies to ensure that no population is overlooked.

## 1. Introduction

Human papillomavirus (HPV) is the most prevalent sexually transmitted virus worldwide, encompassing more than 200 serotypes, with over 40 of them transmitted through direct sexual contact. These can cause anogenital warts or condylomas in both male and female individuals; furthermore, a persistent infection can eventually lead to precancerous and neoplastic lesions affecting the uterine cervix, vulva, and vagina in female individuals, the penis in males, and the anus and oro-pharynx in both sexes [1]. Cervical cancer represents a public health concern, ranking fourth in both incidence and mortality among tumors in women worldwide [2]. After the implementation of the Hepatitis B virus (HBV) vaccination, which has been demonstrated to be capable of reducing the burden of hepatocarcinoma, HPV vaccination is the second vaccine that has shown comparable effectiveness for the prevention of the abovementioned conditions among women and men.

While cervical cancer is a preventable disease, it still represents a global health challenge. In 2020, over 600,000 new cases of cervical cancer were reported, accounting for 6.5% of all cancer diagnoses in women [2]. Over 60% of women with cervical cancer die because of it in low- and middle-income countries, which is twice the proportion in high-income countries, where it stands at around 30% [3]. However, cervical cancer incidence varies significantly within countries, disproportionately impacting ethnic minorities and other marginalized communities [4,5].

### 1.1. Vaccination against the Human Papillomavirus (HPV)

Various vaccines targeting human papillomavirus (HPV) have been developed, approved by the World Health Organization (WHO), and are globally accessible. According to the latest WHO recommendations, six HPV vaccines are indicated to be administered in females aged 9 years and above, and are licensed for use up to 26 or 45 years of age. Some HPV vaccines are also licensed for use in males [6]. All HPV vaccines contain virus-like particles against high-risk HPV types 16 and 18; the quadrivalent vaccine protects also against genotypes 6 and 11, usually related to anogenital warts, while the nonavalent vaccine protects also against genotypes, 31, 33, 45, 52, and 58 [7]. Some changes are expected in the HPV vaccine market with the entry of new manufacturers from China and India [8]. Variation in the utilization of specific vaccines is observed across different regions, influenced by the assessment of locally pertinent data and various factors such as the prevalence of HPV-related health issues (cervical cancer, other HPV-linked cancers, and anogenital warts), the approved target population, product attributes, including single-dose efficacy data, pricing, and program considerations [8,9].

In November 2020, the WHO launched the “Global strategy to accelerate the elimination of cervical cancer as a public health problem”, a guide that highlights some key points to eliminate cervical cancer within a few decades. This document sets out the 90-70-90 targets aiming to fully vaccinate 90% of girls against HPV (i.e., administering the vaccine doses scheduled according to the latest recommendations), to screen for cervical cancer 70% of women by the age of 35 and again by age 45, and to properly and timely treat 90% of women diagnosed with cervical cancer. Reaching these goals by 2030 is expected to help avoid over 62 million cervical cancer deaths by 2120 [10].

In the same year, 2020, the European Center for Disease Prevention and Control (ECDC) released the first guidance on HPV vaccination. Various topics were addressed in this document, including the effectiveness of the nonavalent vaccine. The effectiveness of extending the vaccine to males and the effectiveness of HPV vaccination in people affected by HIV has been demonstrated [1,11].

Adolescent vaccination is a key prevention strategy to decrease the incidence of cervical cancer. The vaccination schedule mostly depends on the age of the recipient. Current WHO guidelines suggest a one- or two-dose vaccine schedule to be fully protected, and an additional dose for special at-risk populations (e.g., people with a compromised immune system; people living with HIV). Furthermore, the WHO recommends also vaccinating boys to curb the virus’ circulation and reduce the incidence of HPV-related lesions [9]. Recent evidence suggests that even a single dose might elicit a durable immune response, thus reducing HPV infections; this could lead to further optimization of vaccination programs in the future [9,12].

After its development, HPV vaccination showed a wider spread in high-income countries than in the rest of the world. Since the HPV vaccine’s introduction in 2006, its global distribution has been uneven. By 2017, more than 100 million adolescent girls worldwide had received at least one dose of the HPV vaccine, and over 270 million doses were administered, with 95% of recipients situated in high-income countries. However, despite these large numbers, achieving global vaccination coverage remains a substantial challenge. As of November 2022, 125 countries (constituting 64% of nations globally) have incorporated the HPV vaccine into their national immunization programs for girls, while 47 countries (24%) have extended the inclusion to boys [13]. However, while the vast majority (90%) of high-income countries have implemented HPV vaccination in their national immunization programs, nearly 60% of lower–middle-income and less than 40% of low-income countries have done so [13].

HPV vaccine availability is unequally distributed across geographic coordinates and income. High vaccine prices, combined with recent supply difficulties, have significantly limited the ability of many countries to introduce the HPV vaccine into their national immunization programs and to ensure the sustainability of current programs [7].

Vaccine hesitancy is currently a recurring theme. It is one of the main concerns in the healthcare sector, precisely because it has direct negative implications on the health of individuals. This is a socially unacceptable phenomenon because it can threaten not only unvaccinated subjects but also the entire population due to the impact on herd immunity [14].

### 1.2. Cervical Cancer in Populations with a Migratory Background

Many factors act as determinants against accessing immunization programs, including coming from a migrant background. A variety of elements can contribute to low vaccination rates and hesitancy to get vaccinated among certain refugee and migrant groups. The decision to accept vaccines is often deeply embedded in the social and historical backdrop, shaped by the individual’s assessment of risk in addition to specific challenges concerning awareness and access to vaccines for some refugee and migrant groups [15]. Currently, one in eight people in the world are a migrant or refugee, and this status can contribute to poor health outcomes; people with a migrant background are often influenced by cultural, personal, social, structural, and economic barriers, all affecting in different ways vaccine access and acceptance. These barriers can only be overcome with the cooperation of the health and non-health actors globally and locally. “The Global strategy to accelerate the elimination of cervical cancer as a public health problem” requires political support from international and local leaders, coordinated cooperation between multisectoral partners, broad support for equitable access in the context of universal health coverage, effective resource mobilization, strengthening of the health system, and vigorous health promotion at all levels [10].

In some migrant and refugee communities, access to clinical health services is often delayed and limited. This evidence has led to growing interest in exploring and identifying the barriers that prevent these populations from accessing healthcare, but also in developing new strategies to increase the knowledge, awareness, and vaccination rates while addressing disparities in cervical cancer incidence and mortality worldwide [16,17].

The objective of this systematic review is to evaluate HPV vaccine adherence among people with migrant and refugee backgrounds globally and investigate its determinants by analyzing barriers to and facilitators of access to prevention measures for HPV.

## 2. Materials and Methods

### 2.1. Literature Search

A systematic review of the scientific literature was conducted following the PRISMA (Preferred Reporting Items for Systematic Review and Meta-Analysis) Statement 2020 guidelines [18]. The systematic review protocol was submitted for registration and accepted in the International Prospective Register of Systematic Reviews (PROSPERO—reference number CRD42024501796).

The criteria for considering studies for the systematic review were based on the population, intervention, comparison, and outcome (PICO) framework:-P (population): the target population of this systematic review included international migrants, refugees (defined according to the definitions provided by the United Nations Convention relating to the Status of Refugees [19]), asylum seekers, regular migrants, migrants in irregular situations, economic migrants, and internally displaced persons.-I (intervention): The intervention of interest was the HPV vaccination as a prevention strategy for cervical cancer. The review focused on studies that evaluate the effectiveness, acceptance, and implementation of HPV vaccination programs within the specified populations.-C (comparison): The comparison involved populations not receiving the HPV vaccine or receiving different cervical cancer prevention strategies. Where possible, we performed before-and-after comparisons within the same population to assess the impact of introducing HPV vaccination programs and comparisons between different migrant groups to assess disparities or differences in vaccination uptake, effectiveness, or outcomes.-O (outcomes): the outcomes of interest included measures related to cervical cancer prevention, such as rates of HPV vaccination uptake; knowledge and attitudes toward HPV vaccination and cervical cancer prevention; barriers to and facilitators of successful vaccination programs.

Searching the PubMed and Scopus databases, the key terms ((transients and migrants) OR (migrants) OR (refugee) OR (nomad)) AND ((uterine cervical neoplasm) OR (cervical cancer) OR (hpv) OR (papillomavirus)) were used for identification of all scientific articles published as of 14 December 2022.

The following inclusion criteria were considered:Peer-reviewed primary studies in English;Studies reporting outcome measures related to cervical cancer prevention strategies, particularly HPV vaccination;Studies that investigated the aforementioned outcomes in the target population.The exclusion criteria were as follows:Publications without an abstract;Articles that were not written in English;Studies whose outcomes were not related to HPV vaccination;Articles in which the target population did not include people with a migratory background and where native populations were also present, the information relating to migrants and/or refugees was not distinguishable;Articles whose study design was a review, systematic review, meta-analysis, trial, or pre–post intervention study;Articles identified as “Commentary”, “Opinion”, “Book”, or “Guidelines”.

In the first phase (“Identification”), duplicates were identified and removed. In the next phase (“Screening”), eight pairs of reviewers selected, based on the title and abstract, only the publications that contained outcome measures related to HPV vaccination. Each pair was assigned the same number of unique publications, and within each pair, the reviewers screened the publications independently. In the third phase (“Quality Assessment”, subsequently abbreviated to “Assessment”), four pairs composed of eight reviewers selected the publications based on the full text. In the screening and assessment phases, an additional reviewer intervened to resolve any disagreements between the reviewers.

### 2.2. Data Extraction and Management

A database created with MS Excel was used for data extraction. For each selected article, the following information was extracted: title, authors, DOI, year of publication, year(s) of study execution, study design (cohort/transversal/cross-sectional/case-control/qualitative study), types of populations (distinguished according to the definitions provided by the United Nations Convention relating to the Status of Refugees) [19], number of people included in the study, number of subjects analyzed, mean and/or median age and/or age range of the study population, study region according to the World Health Organization (WHO) classification [20], countries of origin of the study population, average income range of the country of origin according to the World Bank classification [21], and nationality, ethnicity(ies), and/or religion(s) of the study population.

In this study, geographical classification was based on the regions as defined by the World Health Organization (WHO). The WHO divides the world into six regions, namely, the African Region (AFR), covering the majority of the African continent; the Region of the Americas (AMR), encompassing all of the countries in North, South, and Central America, as well as the Caribbean; the South-East Asia Region (SEAR), including countries in the South-East Asian part of the continent; the European Region (EUR), covering a broad geographic area from the Atlantic to the Pacific; the Eastern Mediterranean Region (EMR), including countries in North Africa, the Middle East, and Western Asia; and the Western Pacific Region (WPR), which is made up of countries in East Asia and the Pacific. 

Regarding the outcome measures related to HPV vaccination, the following information was retrieved: number of people who could be vaccinated, number of people vaccinated with a complete or partial schedule, number of people aware of the availability or benefits of vaccination, and number of people aware of the risks of HPV infection. Where possible, the data were separated based on the sex of the subjects.

To build a population profile, additional information was collected, including level of education, according to the International Standard Classification of Education (ISCED) [22], number of people able to understand the local language or a language commonly used in welfare services, number of people married or in a civil union or cohabiting with a partner, number of people who are unmarried, widowed, divorced, separated or in any case not cohabiting with a partner, time spent in the host country, time spent in the host country until first access to local health services, annual income, number of people with paid work, and number of people vaccinated against other infectious diseases. Furthermore, barriers to and facilitators of access to health services were investigated.

In quantitative studies in which the eligible and vaccinated population could be calculated, those with a statistically significant association with vaccination outcome were identified as barrier or facilitating factors. For qualitative studies (e.g., focus groups and open interviews), however, barriers or facilitators were identified and reported by study participant themselves. Figure 1 shows the selection phases of our systematic review. The literature search returned 273 records on PubMed, and 318 records on Scopus; after the exclusion of 129 duplicates, 462 articles were subjected to the screening phase. A total of 309 articles passed the screening phase; 4 reports, whose full text was not retrieved, were eventually excluded. Therefore, 305 articles were subjected to the assessment phase, at the end of which 34 studies were identified regarding the HPV vaccination and/or the determinants to its access. The Newcastle–Ottawa assessment scale (NOS-scale) was used for assessing the methodological quality of quantitative studies. This tool contains two forms, one adapted for cross-sectional studies and one for cohort studies (Appendix A).

If all criteria of methodological quality were fulfilled within the domains, points were assigned to the respective study. The NOS-scale was adapted for the purpose of this review and cross-sectional studies could receive a maximum of 10 points, while a cohort study could receive a maximum of 9 points. The results of this process are available in Appendix B.

### 2.3. Data Analysis 

We did not carry out a meta-analysis due to the heterogeneity of the included studies. The findings of this review are presented as a descriptive synthesis.

## 3. Results

The studies included in our review are shown in Table 1. Seventeen of these studies included quantitative data on the adherence to vaccination; the remaining seventeen articles were merely qualitative studies. Twenty-two (64.7%) of the selected studies were conducted in the Region of the Americas (AMR), nine (26.5%) were conducted in the European Region (EUR), two (5.8%) studies in the Western Pacific Region (WPR), and one study (2.9%) in the South-East Asian Region (SEAR).

From the selected studies, the following data emerged regarding the adherence to HPV vaccination: the average adherence pooled rate was 34.5%; stratifying by sex, the average adherence was 44.4% in females pooled from multiple studies who had completed the vaccination schedule, while for six studies, which included females who had not completed the pooled rate, it was 17.4%. One study investigated vaccine adherence in men, with a percentage of adherence to the complete schedule of 0.6%. Based on data from two studies, however, 3% was the pooled rate for males who did not complete the entire vaccination cycle. The pooled rate for vaccination compliance with the complete schedule in 14 studies that considered females and/or males was 34.5%. Eight studies evaluated adherence regardless of the sex and the vaccination initiation pooled rate was 31.6% (Table 2).

When stratified by the WHO region of the study, the pooled adherence rate to the complete vaccination cycle was 63.4% based on the results of six studies conducted in Europe ([27,28,44,45,53,55]), and 6% according to nine studies conducted in the AMR. According to the only study carried out in the EUR, people’s vaccine initiation rate was 20.8% [53], whereas according to five studies conducted in the AMR, the pooled rate was 32.8% [43,47,48,49,50,51,54]. Further stratifying by sex, relevant differences appeared in the results of the studies conducted in the AMR: the results of seven studies showed a 13.3% pooled adherence rate to the complete vaccination schedule [23,45,46,47,48,49,50], while the pooled initiation rate was 16.1% [47,48,49,50,51].

Stratifying by migratory status, the results of 13 studies show a pooled rate of 44.7% adherence for international migrants [27,28,29,43,44,45,47,48,49,50,52,53,55], and according to 4 studies, refugees’ pooled adherence rate was 0.6% [25,51,54,56]. On the other hand, the initiation pooled rates were, respectively, 14.9 [43,47,48,49,50,53] and 48% [51,54]. Further stratifying by sex, female international migrants’ pooled adherence rate was 45.4% [27,29,43,44,45,47,48,49,50,52,53,55], while for female refugees, it was 27.7% [25,51]. The initiation pooled rate for female international migrants and refugees was 18.0% [47,48,49,50,53] and 13.8% [51], respectively.

Table 3 shows the identified barriers and facilitators, in both qualitative and quantitative studies.

As shown in Table 3, some barriers have been highlighted in several studies; for example, “Lack of Health knowledge/literacy; Lack of promotion programs; Lack of motivation” occurred in twenty studies (58.8%). The “Lack of trust in health workers; lack of regular health check-ups” and “Low socio-economic level” were recurring barriers in nine (26.5%) and twelve (35.3%) studies, respectively. In five studies (14.7%), however, “Perception of vaccine-related risks” was highlighted as an obstacle. On the other hand, the most common facilitator appeared to be “Adequate communication between patients and health workers; Regular access to health services or a specific type of doctor (general practitioner (GP), pediatrician or other specialists)”, highlighted in fifteen articles (44.1%) as a relevant factor encouraging vaccination acceptance and adherence. The second most common facilitator, recurring in eleven studies (32.3%), was “Increasing awareness of prevention strategies; People empowerment; Increasing health literacy; Information or promotion programs”.

## 4. Discussion

The data reveal that migratory status significantly influences participation in vaccination programs. However, it is notable that few studies categorize the participants as “refugees”. As outlined in Table 3, it is widely understood that several factors are involved in determining vaccine uptake, serving as either barriers or facilitators. This underscores the complexity of vaccination adherence within migrant populations, suggesting the need for multifaceted approaches in public health strategies.

### 4.1. Health Information/Health Literacy/Motivation

In our review, the most recurring factor influencing lower HPV vaccine uptake appeared to be the lack of adequate health information or literacy. This highlights the crucial role of accessible and comprehensible health education in promoting vaccine programs. Several studies reported, indeed, how the participants were lacking information on cervical cancer [23,24,25], often unaware of the importance of vaccines in preventing cervical cancer [26,27,28,29,30], or even that such a vaccine existed and was available [23,24,28,31,32,33,34,35,36]. 

Some studies reported that people had low or no knowledge of HPV or other sexually transmitted infections (STIs) [24,28,29,32,37,38], hence the low acceptance of the vaccine. Particularly, one study reported parents’ hesitancy toward vaccinating their sons, though aware of the risk of cervical carcinoma related to HPV infection, as they could not understand “how does that have to do with boys” [24]. This element shows the importance of a complete health education delivered by health professionals. 

The evidence shows that the effects of interventions to increase health literacy and awareness of the benefits of HPV vaccination were less evaluated, yet they recurred as the second most relevant determinant of vaccine acceptance. Indeed, some studies highlighted the parents’ willingness to vaccinate their own children after receiving adequate information on HPV [26,31,32], the HPV vaccine [24,29,30,31,38], or on how to get vaccinated themselves if they had the chance [33,40]. In some studies, the participants expressed their curiosity about the HPV vaccine and its characteristics, such as the composition, the protection it offers, potential side effects, and related details [25,29,30,38]. In these studies, people suggested ways to provide such information, for instance, educational workshops to raise awareness about health topics among the youth [25,30], or information given by the provider, in both oral and written form, so people better retain the acquired knowledge [29]. Some others preferred to receive health information through digital media [25,29,40,55]. In any case, people seemed to express the need for a more user-friendly approach. Indeed, the lack of knowledge appeared to be attributable to the lack of official information by national or local health authorities [30,40,41,42], or recommendations by professionals [26,38]. In some cases, vaccination adherence was affected by both low health literacy and a lack of personal interest and motivation due to concomitant health problems, leading people to prioritize these and postpone the vaccination [43].

### 4.2. Trust and Communication with Health Professionals; Regular Access to Health Services

Another recurring factor that negatively affects access to vaccinations is the mistrust in the health system of the host country or in local health professionals. People recruited in one study reported their concern about the vaccine providers being adequately prepared; they believed that neither GPs nor nurses had received adequate training to visit their children and evaluate if they could get vaccinated in the presence of any illness symptoms [40]. In one other study, people preferred to postpone vaccinations for their children so they could be administered in their country of origin [44]. In one study, mistrust in the pediatrician was reported, who was “not doing the correct follow up” [30]. In some cases, people associated their own mistrust in the health system with the locals’ alleged dependence on medications and/or vaccines and the perception of a higher prevalence of illnesses. Therefore, they could not trust the efficacy of the vaccine on their healthy children [29]. Someone simply ascribed their non-adherence to the lack of providers’ recommendations [26,33] or inadequate communication [29,38]. In one study, a participant felt more comfortable in sharing their opinions with other parents because the physician’s recommendations appeared quite unclear to her; in some cases, this might result in vaccine hesitancy or, instead, in a clarification of the recommendations and a higher acceptance [29]. Some people, instead, reported trust in their physicians’ or other health professionals’ opinions [24,25,30,33,37,40,42,45], although, as mentioned previously, they declared that they preferred written advice to help memorize it. Some studies also highlighted how people would appreciate receiving phone calls, text messages, or letters from the vaccination providers as a means to promote vaccine uptake [25] or as reminders about scheduled appointments [30]. In one of the studies, the provision of complete information was considered an essential responsibility of the healthcare providers in order to give the migrant parents the chance to make wiser decisions [29]. Some individuals considered themselves to be the most influential persons in making their own health decisions, and therefore, they needed to acquire information on prevention benefits [25,33,38]. Also, a lack of regular health check-ups because of perceived unnecessity was reported as a determinant of low access to vaccination [46]. Conversely, several studies reported how having a usual source of care or a higher frequency of visits with a healthcare provider was associated with an increase in vaccination adherence [39,47,48,49,50,51].

### 4.3. Socio-Economic Level

Several studies reported an association between low socio-economic level and low adherence to preventive measures [23,27,30,31,33,37,40,44,52,53]. In this context, the term “socio-economic level” refers to a complex of determinants including formal education level, family income, housing conditions, etc. In some cases, a statistically significant correlation was found with low education level [23,31,53]; this may eventually result in lower proficiency of the host language and lower health literacy. Furthermore, it is widely acknowledged that HPV vaccination acceptance is directly influenced by the level of health literacy, the perception of severity, susceptibility to disease, and vaccination barriers and benefits [57]. Another recurring determinant was a low family income, which negatively affected vaccination uptake [31,37,53], and in certain studies, participants were less inclined to receive the vaccine if they had to pay for it [27,30,31,33,40,54,55]. Indeed, studies reported how free offers of the vaccination would improve its acceptability [23,30]. On the other hand, participants in one of the studies declared that they were willing to be vaccinated even if they had to pay for it, if the price was affordable [32], because it meant making an investment in safeguarding their own health. In some cases, a lack of transportation affected the possibility of reaching vaccination facilities, and to overcome this barrier, people suggested offering the vaccination in the same clinic where they receive clinical consultations [30]. 

### 4.4. Family and Community Support

Another determinant of low vaccination uptake was a lack of family support; women reported that they had to take care of their children’s health without their husbands’ help, while dealing with other family issues or their own job [30]. In some studies, a statistically significant correlation was found between the parents’ marital status and vaccination initiation or completion, implying that a lack of family support led to the prioritization of some other aspects of family life over prevention services [31,47,53]. Indeed, some women reported receiving support and underlined the importance of their husbands’ opinion on their children receiving vaccinations [25,29,30]. In one case, being unmarried was related with the misconception that gynecological examination was necessary only for married and sexually active women, not for those unmarried, and that being visited by a gynecologist for a single woman was associated with socially unacceptable promiscuity [41]. 

Another highlighted factor was the lack of flexibility at work, such as not having permission to leave work to attend medical appointments either for them or for their children. Therefore, vaccine administration in the community, at school, or through mobile clinics was suggested as a facilitator [30]. Finally, another strategy that was identified to overcome this barrier was the possible inclusion of employers in the development of preventive health programs [30].

### 4.5. Administrative Factors

In some cases, there were factors related to health services themselves hindering the adherence to a vaccination schedule. For instance, people reported difficulties in negotiating appointments that would not interfere with children’s school schedules, or having to wait long periods before the appointment, sometimes also facing a lack of vaccines in the facilities [30]. It was suggested that if the clinic had been in geographic proximity or had been accessible in the afternoon and evenings, these would have facilitated the enhancement in vaccine uptake [30].

### 4.6. Language

In a close correlation with the previous factors, “language barriers” were considered responsible for migrants’ and refugees’ lower adherence to immunization programs, and in some cases, it hindered already their access to GP care [40]. A lack of fluency in the host country language was also related to inadequate health literacy and difficulties in navigating the healthcare system [30,41], therefore resulting in ignorance of the preventive services. Language barriers were also dependent on the education level, yet even for people who had accessed higher education, their comprehension of medical terminology in English was limited, and migrants and refugees expressed difficulties in describing their symptoms in a different language [40,41]. Participants of some studies underlined their need to interact with health professionals who spoke the same language as a facilitator for building trust [24], to deliver educational messages and spread awareness [25], and in general to facilitate better access to healthcare [30].

### 4.7. Ethnicity and Cultural Factors

Throughout several studies, a difference emerged in vaccination uptakes between locals and migrant and refugee people [27,44,46,47,53,56]. In some cases, the lack of knowledge about the vaccine among these population groups was attributable to the unavailability of the vaccine in their country of origin [54,56]. Some countries did not include this HPV vaccination in their immunization programs, while on the other hand, some health systems did not include migrants in their vaccination campaigns, which did not consider the peculiar needs of these populations and obstacles such as language barriers or working and family conditions. 

Cultural believes and attitudes were reported to undermine trust in health professionals; for example, in one study, some women declared having felt uncomfortable in the presence of their doctors because of traditional gestures that were misinterpreted as a lack of engagement in healthcare conversations [24]. Therefore, the presence of trained healthcare staff, as well as interpreters or cultural mediators, or staff sharing origins with the health facility’s users can be of help to overcome this kind of barrier [24,30].

At other times, vaccine uptake willingness was hindered by sex-related misconceptions and taboos. In some studies, it emerged that discussing sex was considered unacceptable, and parents declared not being willing to vaccinate their children, especially girls, because it would promote a promiscuous sexual attitude [29,30,33,54,56]. In one study, high levels of acculturation were mentioned as facilitating factors of vaccine acceptancy [48]. Overall, proper health communication skills appear to be crucial in overcoming this kind of barrier to educate people to prioritize their children’s and their own health.

### 4.8. Perception of Risks and Benefits

In some studies, mistrust of vaccines emerged as a deterrent from immunization. Some people feared and reported bad experiences with vaccination side effects [24,29,30,40,41] or simply feared the needle itself [30]. This was likely related to the abovementioned lack of knowledge and health literacy. Indeed, when properly informed about side effects and, especially the benefits of being vaccinated, people appear to be willing to get their children vaccinated [30,32,38].

### 4.9. Limitations

This systematic review’s broad approach aimed at assessing HPV vaccination rates and identifying barriers or facilitators among diverse migrant groups may lead to a loss of specific insights due to its wide focus. This can result in a loss of complex and nuanced differences in vaccination impact and uptake among the different categories of migrants as well as contexts, making it challenging to offer targeted policy recommendations. 

Challenges were also encountered due to the varying quality of the included studies, such as inconsistent definitions of migrant populations used by the different authors and outcomes related to HPV vaccination, which complicated the comparison and synthesis of the findings. Additionally, the variability in the design of the studies included in the review and a predominance of cross-sectional data shifted the direction of causality between barriers and/or facilitators and HPV vaccine uptake, highlighting the need for more longitudinal studies to understand the long-term impact of HPV vaccination programs on migrant populations. Finally, most of the studies included in this systematic review were conducted in high-income countries, and therefore, several significant research gaps exist in the current evidence in low- and middle-income countries.

## 5. Conclusions

Many of the barriers and facilitators listed above are highly interconnected, influencing each other and eventually impacting people’s health-seeking behaviors. One factor that often negatively affects access to vaccination programs is a low socio-economic level and people with a migratory background are commonly affected by this factor. 

The adaptation process to the host community pushes migrants and refugees to prioritize the search for better housing, working, and living conditions, putting aside their own health. However, the factor that appeared to be reported the most was health literacy; a lack of information on the risks of STIs, on ways to prevent them, and eventually on the benefits of adhering to prevention programs determines a low vaccine acceptance. This is often associated with a low frequency of health check-ups and contact with healthcare providers. Building trust in health services and healthcare personnel is highly needed to provide quality of care and give people the possibility to make proper decisions regarding their own health. 

In addition, when interacting with migrant and refugee populations, cultural factors may intervene, and therefore, proper education of healthcare professionals is necessary for them to be able to tackle cultural barriers hindering access to preventive measures. Furthermore, language barriers often limit the trust-building mechanisms and affect the capability to receive complete and clear information; including interpreters, cultural mediators, and multi-language communication tools and media would be of great help in engaging with people with a migratory background. It is necessary to take all of these determinants into account when developing health promotion strategies and prevention plans to leave no one behind and guarantee adequate levels of healthcare to everyone. 

HPV vaccination rates can be improved only by trying to address the specific challenges in each country, at the structural, cultural, and economic level. This is why multi-sectoral interventions are needed that are tailored to the community and culturally and linguistically appropriate. 

It must be noted that most of the studies included in this systematic review were conducted in two WHO regions (EUR and AMR) and particularly in high-income countries; therefore, information on vaccine acceptancy and adherence, and the factors influencing them, in low- and middle-income countries is still lacking. Countries with lower incomes might indeed be unable to sustain the costs of scientific research; it is necessary to promote and sustain the research, to collect proper and valuable data, and to implement adequate health promotion policies according to scientific evidence.

## Figures and Tables

**Figure 1 vaccines-12-00256-f001:**
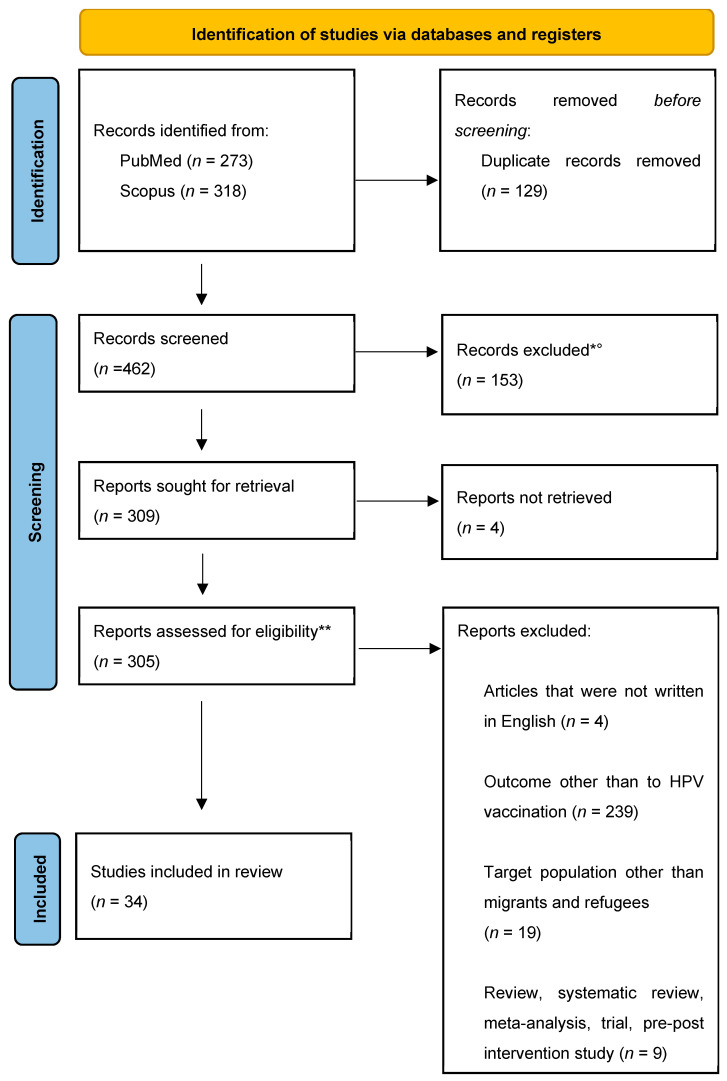
PRISMA 2020 flow diagram for new systematic reviews. * Eight pairs of human reviewers selected, based on the title and abstract, only the publications that contained outcome measures related to HPV vaccination. ° Abstract not available (*n* = 16); language of the study was not English (*n* = 7); target population other than migrants and refugees (*n* = 21); outcome not related to HPV vaccination (*n* = 28); review, systematic review, meta-analysis, trial, and pre–post intervention study (*n* = 81). ** Four pairs of human reviewers selected the publications on the basis of the full text [18].

**Table 1 vaccines-12-00256-t001:** Characteristics of included literature (*n* = 34).

Authors (Date) [Reference]	Study Design	Newcastle–Ottawa Scale Cross-Sectional Studies Total (Out of 10)	Newcastle–Ottawa Scale Cohort Studies Total (Out of 9)	WHO Region	Population (N)	*%* of Vaccinated People (for Quantitative Studies)
Bhatta et al. (2020) [23]	Cross-Sectional	4	-	SEAR	Refugees (90)	N/A
Ghebrendrias et al. (2021) [24]	Qualitative	-	-	AMR	Refugees (18)	N/A
Allen et al. (2019) [25]	Qualitative	-	-	AMR	Refugees (31)	22.6
Aragones et al. (2016) [26]	Qualitative	-	-	AMR	International Migrants (36)	N/A
Pollock et al. (2019) [27]	Cross-Sectional	4	-	EUR	International Migrants (1172)	71.6
Napolitano et al. (2018) [28]	Cross-Sectional	5	-	EUR	International Migrants (42)	0.7
Ko et al. (2019) [29]	Qualitative	-	-	AMR	International Migrants (30)	20.0
Vamos et al. (2021) [30]	Qualitative	-	-	AMR	International Migrants (13)	N/A
Lin et al. (2020) [31]	Cross-Sectional	8	-	WPR	Internal Migrants (7059)	N/A
Khodadadi et al. (2021) [32]	Cross-Sectional	3	-	AMR	International Migrants (313)	N/A
McComb et al. (2018) [33]	Qualitative	-	-	AMR	International Migrants (11)	N/A
Kim et al. (2015) [34]	Qualitative	-	-	AMR	International Migrants (12)	N/A
Wilson et al. (2021) [35]	Qualitative	-	-	AMR	International Migrants (41)	N/A
Ganczak et al. (2021) [36]	Qualitative	-	-	EUR	International Migrants (22)	N/A
Lindsay et al. (2020) [37]	Cross-Sectional	3	-	AMR	International Migrants (54)	N/A
Dailey et al. (2015) [38]	Qualitative	-	-	AMR	International Migrants (20)	N/A
Seo et al. (2017) [39]	Qualitative	-	-	AMR	International Migrants (12)	N/A
Gorman et al. (2019) [40]	Qualitative	-	-	EUR	International Migrants (13)	N/A
Lee et al. (2017) [41]	Qualitative	-	-	AMR	International Migrants (16)	N/A
Burke et al. (2015) [42]	Qualitative	-	-	AMR	Refugees (25)	N/A
Mohareb et al. (2021) [43]	Cohort	-	5	AMR	International Migrants (34)	2.9
Hertzum-Larsen et al. (2020) [44]	Cohort	-	7	EUR	International Migrants (5990)	71.0
Remschmidt et al. (2014) [45]	Cross-Sectional	4	-	EUR	International Migrants (286)	51.0
Patel et al. (2020) [46]	Cross-Sectional	4	-	EUR	International Migrants (82)	N/A
Pérez et al. (2017) [47]	Cross-Sectional	9	-	AMR	International Migrants (7379)	3.7
Cofie et al. (2018) [48]	Cross-Sectional	8	-	AMR	International Migrants (3080)	8.1
Beltran et al. (2016) [49]	Cross-Sectional	5	-	AMR	International Migrants (192)	16.7
Pruitt et al. (2015) [50]	Cross-Sectional	5	-	AMR	International Migrants (248)	23.0
Kenny et al. (2021) [51]	Cross-Sectional	5	-	AMR	Refugees (65)	27.7
Kamimura et al. (2015) [52]	Cross-Sectional	4	-	AMR	International Migrants (88)	9.1
Slåttelid Schreiber et al. (2015) [53]	Cohort	-	9	EUR	International Migrants (1522)	63.6
Berman et al. (2017) [54]	Cross-Sectional	6	-	AMR	Refugees (2269)	0
Marques et al. (2022) [55]	Cross-Sectional	7	-	EUR	International Migrants (1100)	14.8
Nyanchoga et al. (2021) [56]	Cross-Sectional	6	-	WPR	Refugees (77)	7.8

**Table 2 vaccines-12-00256-t002:** Adherence to HPV vaccination by sex, region of the study, and migration status.

		% (95%CI)	N. of Studies
Stratified by sex	People with complete schedule	34.5 (27.4–41.5)	17 [25,27,28,29,43,44,45,47,48,49,50,51,52,53,54,55,56]
	People with incomplete schedule	31.6 (22.3–40.9)	8 [43,47,48,49,50,51,52,53,54]
	Female with complete schedule	44.4 (28.7–60.2)	12 [25,27,44,45,47,48,49,50,51,52,53,55]
	Female with incomplete schedule	17.4 (11.9–22.9)	6 [47,48,49,50,51,52,53]
	Male with complete schedule	0.6 (0.3–0.9)	1 [47]
	Male with incomplete schedule	3.0 (2.4–3.6)	2 [47,49]
Stratified by sex and region of the study	People in EUR (complete schedule)	63.4 (48.0–78.8)	6 [27,28,44,45,53,55]
	People in AMR (complete schedule)	6.0 (3.9–8.2)	9 [25,29,43,47,48,49,50,51,52]
	People in EUR (incomplete schedule)	20.8 (18.8–23.0)	1 [53]
	People in AMR (incomplete schedule)	32.8 (22.5–43.1)	7 [43,47,48,49,50,51,54]
	Female in EUR (complete schedule)	63.4 (48.0–78.8)	5 [27,44,45,53,55]
	Female in AMR (complete schedule)	13.3 (9.7–17.0)	7 [25,47,48,49,50,51,52]
	Female in EUR (incomplete schedule)	20.8 (18.8–23.0)	1 [53]
	Female in AMR (incomplete schedule)	16.1 (10.9–21.3)	5 [47,48,49,50,51]
	People in WPR (complete schedule)	7.8 (7.09–8.52)	1 [56]
Stratified by sex and migration status	International Migrants (complete schedule)	44.7 (28.4–61.0)	13 [27,28,29,43,44,45,47,48,49,50,52,53,55]
	Refugees (complete schedule)	0.6 (0.6–1.8)	4 [25,51,54,56]
	International Migrants (incomplete schedule)	14.9 (10.3–19.6)	5 [43,47,48,49,50,53]
	Refugees (incomplete schedule)	48.0 (37.5–58.5)	2 [51,54]
	Female International Migrants (complete schedule)	45.4 (29.2–61.6)	12 [27,29,43,44,45,47,48,49,50,52,53,55]
	Female Refugees (complete schedule)	27.7 (17.3–40.2)	2 [25,51]
	Female International Migrants (incomplete schedule)	18.0 (12.0–23.9)	5 [47,48,49,50,53]
	Female Refugees (incomplete schedule)	13.8 (6.5–24.7)	1 [51]

**Table 3 vaccines-12-00256-t003:** Barriers to and facilitators for accessing the HPV vaccination in migrant populations.

Barriers	N. of Studies	Facilitators	No. of Studies
Lack of health knowledge/literacy; lack of promotion programs; lack of motivation	20 [23,24,25,26,27,28,29,30,31,32,33,34,35,36,37,38,40,41,42,43]	Increasing awareness of prevention strategies; people empowerment; increasing health literacy; information or promotion programs	11 [24,25,26,29,30,31,32,33,38,40,55]
Lack of trust in health workers; lack of regular health check-ups	9 [26,29,30,33,38,40,44,45,46]	Adequate communication between patients and health workers; regular access to health services or a specific type of doctor (general practitioner (GP), pediatrician, or other specialists)	16 [24,25,29,30,33,37,38,39,40,42,44,47,48,49,50,51]
Low socio-economic level	12 [23,27,30,31,33,37,40,44,52,53,54,55]	Offer the service at the workplace or in the community	2 [23,30]
Lack of family or social support	4 [30,31,47,53]	Family, friends, or community support	3 [25,29,30]
Administrative factors (inadequate distribution of vaccination facilities, long waiting lists, and long queues at the centers)	1 [30]	Administrative factors (adequate distributions of the facilities and free access days)	1 [30]
Language barriers	3 [30,40,41]	Information translation; language interpreting	3 [24,25,30]
Ethnicity/origin; cultural factors	11 [24,27,29,30,33,44,46,47,53,54,56]	Cultural mediation; transcultural training	3 [24,30,48]
Perception of vaccine-related risks	5 [24,29,30,40,41]	Awareness of the benefits of vaccination	3 [30,32,38]

## Data Availability

The original contributions presented in the study are included in the article, further inquiries can be directed to the corresponding author.

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
