# Peer review of "Barriers to and Facilitators for Accessing HPV Vaccination in Migrant and Refugee Populations: A Systematic Review"

_vaccines, 2024, doi:10.3390/vaccines12030256_

Round 1

Reviewer 1 Report

Comments and Suggestions for Authors

The authors provide a detailed overview of HPV vaccine adherence and their facilitators and barriers in immigrant populations, stratified by gender, WHO region, and immigration status. This study includes a total of 34 studies, of which 17 were quantitative and the remainder were qualitative. Low socioeconomic status and individuals' health literacy were identified as influencing factors.

1. It seems that the abstract does not reflect the results of HPV vaccine adherence.

2. The study population is confusing. The manuscript title refers to immigrant populations, the abstract and various parts of the manuscript mention immigrants and refugees, yet it seems that the manuscript does not provide a clear definition of the study population.

3. The quality assessment should describe the evaluation tools, and the assessment results is suggested to be placed in the supplementary materials.

4. In line 164 and 174, the age and education level of the study population were extracted, but why were these not included in the statistical analysis? Occupation also seems to influence HPV vaccine adherence; consider including these factors in the study.

5. The classification of geographical areas from lines 208-215 should be placed in the methodology section. Lines 216-219 should not simply describe the quantity of research, and rates in different geographical areas should be combined. It is suggested to visualize the pooled rates on a world map to show the differences.

6. Figure 1 seems incomplete.

7. All tables should be in the form of a three-line table.

8. Some rates in the manuscript are pooled from multiple studies, while others are obtained from a single study. This should be distinguished in the text, for instance, rates pooled from multiple studies are referred to as the pooled rate.

9. Line 244, the number of decimal places should be consistent.

10. In Table 3, language, ethnicity, and culture are both considered barriers and facilitators. This may confuse the readers, so the author should differentiate between them.

Reviewer 2 Report

Comments and Suggestions for Authors

Davide Graci et al. have undertaken a systematic review to assess HPV vaccine uptake among migrant and refugee populations and to identify factors influencing this uptake. Cervical cancer, caused by persistent infection with the human papillomavirus (HPV), is a significant global health concern, addressable through available HPV vaccines. This review, conducted in accordance with the PRISMA Statement 2020 guidelines, analyzed thirty-four studies, which included both quantitative data on HPV vaccine uptake and qualitative evaluations. The findings point to low socio-economic status as a barrier to vaccine access. Notably, health literacy emerged as a critical factor, with a gap in awareness about STIs, their prevention, and the benefits of vaccination. Language barriers were also seen as obstacles that impede trust-building and the effective communication of health information. The authors highlight the need to consider these factors in devising effective public health strategies, aiming to ensure that prevention plans are inclusive of migrant and refugee populations.

The claims are properly placed in the context of the previous literature. The experimental data support the claims. The manuscript is written clearly enough that most of it is understandable to non-specialists. The authors have provided adequate proof for their claims, without overselling them. The authors have treated the previous literature fairly. The paper offers enough details of methodology so that the experiments could be reproduced.

Comments

1. In this review, the authors aim to identify factors influencing vaccine uptake and barriers to vaccine access. However, I would like to comment on the terminology used. The term 'anti-HPV vaccination' is employed throughout the manuscript. I recommend using more commonly accepted formulations such as 'HPV vaccine uptake' or 'HPV vaccination.' These terms are widely recognized and would enhance the clarity and precision of the manuscript.

2.
The manuscript states that cervical cancer is 'predominantly caused by HPV.' However, this phrasing may not fully capture the causal relationship between HPV and cervical cancer. It's important to note that cervical cancer is directly caused by persistent infections with specific types of human papillomavirus (HPV). I recommend revising this statement to reflect the direct and established link between HPV and cervical cancer more accurately.

3. In the manuscript's introduction, it is mentioned that 'Current WHO guidelines recommend two or three doses of the vaccine for females and three doses for those with compromised immune systems.' It is important to note that the WHO updated its HPV vaccination recommendations in April 2022. The new guidance suggests a one or two-dose schedule for certain age groups. I recommend revising this section to align with the latest WHO recommendations, ensuring the manuscript reflects the most current public health guidelines.

https://www.who.int/news/item/20-12-2022-WHO-updates-recommendations-on-HPV-vaccination-schedule

4.
In the introduction, the manuscript states that 'Between 2006, when the first HPV vaccine was licensed, and 2017, over 100 million adolescent girls worldwide received at least one dose of the HPV vaccine, 95% of whom were in high-income countries.' It should be updated to include the broader context of HPV vaccine distribution and monitoring. As of 2017, over 270 million doses have been administered globally, including 120 million in the US alone. Moreover, the global vaccination rate remains a challenge, with only one in eight girls receiving the vaccine worldwide. This broader perspective on both the distribution and the ongoing safety monitoring of the HPV vaccine would provide a more comprehensive view of its global impact.

https://www.cancer.org/cancer/risk-prevention/hpv/hpv-vaccine-facts-and-fears.html

https://prescancerpanel.cancer.gov/report/hpvupdate/HPVCancers.html

https://www.unicef.org/supply/stories/closing-gap-unicef-bolsters-country-efforts-increase-hpv-vaccination

5. In the introduction, it is stated that 'These lesions represent the first type of cancer which is preventable through vaccinations among women and men.' However, it's important to note that the Hepatitis B vaccine, which can prevent hepatocellular carcinoma, was introduced before the HPV vaccine. Thus, the Hepatitis B vaccine was actually the first to offer cancer prevention. I suggest revising this statement to correctly acknowledge the precedence of the Hepatitis B vaccine in cancer prevention through vaccination.

6. In the introduction, the manuscript states: 'Some types of HPV can cause genital warts or condylomas, while others can lead to dysplasia, including cancers of the uterine cervix, vulva, vagina, penis, anus, and oro-pharynx.' However, this wording may imply that all these anatomical areas are relevant to both genders, which is not accurate. Please consider revising this sentence to accurately reflect that certain HPV-related cancers affect specific genders, such as cervical, vulval, and vaginal cancers predominantly in women, and penile cancer in men.

7. In the introduction, the authors state that 'Several vaccines against HPV have been developed, which are currently available and authorized both in the United States and in Europe.' It should be noted, however, that in the United States, only the nonavalent HPV vaccine is currently available, as the bivalent and quadrivalent vaccines have been withdrawn from the market. Additionally, while China has developed its own HPV vaccines, these are not authorized or available in the United States or Europe. This distinction is important for the accuracy of the manuscript and the understanding of global HPV vaccine availability.

8. Many discussions on HPV vaccination center on informing parents and adolescents about the decision to vaccinate. I believe that framing HPV vaccination as a significant personal choice linked to future sexual behavior may not be necessary. In countries where vaccines included in the childhood immunization program have higher coverage rates than the HPV vaccine, integrating HPV vaccination into these existing programs might simplify the process. This approach could reduce the need for extensive information campaigns aimed at parents, positioning the HPV vaccine as a standard, routine immunization for children, similar to other recommended vaccines.

Minor revisions

Line 2-3, title, "Understanding HPV Vaccine Uptake Among Migrant Populations: A Systematic Review of Barriers and Facilitators"

Line 14-28, abstract, "Human papillomavirus (HPV), a primary cause of cervical cancer, is the most prevalent sexually transmitted virus worldwide. Despite the availability of effective vaccines, HPV vaccination rates vary, especially among migrant and refugee populations. This systematic review, adhering to PRISMA 2020 guidelines, examines 34 studies to evaluate HPV vaccine uptake in these groups and identify influencing factors. Key determinants include socio-economic status and health literacy, with language barriers impacting information access and trust. Understanding these factors is vital for developing inclusive vaccination strategies to ensure no population is overlooked."

Line 33-37, introduction, "Human papillomavirus (HPV) is a globally prevalent sexually transmitted virus, encompassing over 200 serotypes, with more than 40 transmitted through direct sexual contact. These serotypes can lead to a range of conditions from genital warts to cancers affecting the uterine cervix, vulva, vagina, penis, anus, and oro-pharynx. Cervical cancer, in particular, is a significant public health concern, as it is directly caused by persistent HPV infections and ranks fourth worldwide in both incidence and mortality rates."

Line 41-47, introduction, "While cervical cancer is preventable, it remains a significant global health challenge. In 2020, over 600,000 new cases were reported, accounting for 6.5% of all female cancer diagnoses. The mortality rate is strikingly high in low- and middle-income countries, exceeding 60%, which is more than double the rate in high-income countries, where it stands at around 30%. This disparity is not just a matter of geography; even in affluent nations, cervical cancer incidence varies significantly, disproportionately impacting ethnic minorities and marginalized communities."

Line 49-54, introduction, "Various HPV vaccines have been developed and authorized, with differing availability across regions. In the United States, the nonavalent vaccine, which protects against genotypes 6, 11, 16, 18, 31, 33, 45, 52, and 58, is predominantly used. These genotypes are associated with the majority of cervical and other HPV-related cancers. Notably, the bivalent and quadrivalent vaccines are no longer available in the U.S. In contrast, the availability and types of HPV vaccines can vary in Europe and other regions. The widespread use of these vaccines is primarily seen in high-income countries."

Line 60-67, introduction, "Adolescent vaccination is a key primary prevention strategy to decrease cervical cancer incidence. The WHO's current recommendations, updated in April 2022, suggest a one or two-dose HPV vaccine schedule for specific age groups, with additional doses advised for those with compromised immune systems. Moreover, WHO endorses vaccinating boys to curb virus transmission and reduce HPV-related lesions. Emerging evidence indicates that even a single vaccine dose might elicit a durable immune response and help lower HPV infection rates, which could lead to further optimization of vaccination programs in the future."

Line 77-84, introduction, "Since the HPV vaccine's introduction in 2006, its global distribution has been uneven. By 2017, over 270 million doses had been administered worldwide, with a significant number, including 120 million, in the United States. However, despite these large numbers, the global vaccination rate remains a challenge, with only one in eight girls vaccinated against HPV. This disparity is particularly evident when comparing high-income countries, where nearly 90% have implemented the vaccine in national programs, to less than 40% in low-income and under 60% in lower-middle-income countries."

https://www.unicef.org/supply/stories/closing-gap-unicef-bolsters-country-efforts-increase-hpv-vaccination

Line 262-266, discussion, "The data reveals that migratory status significantly influences participation in vaccination programs, though it's notable that few studies specifically categorize participants as 'refugees.' As outlined in Table 3, various factors contribute to vaccine uptake, serving either as barriers or facilitators. This underscores the complexity of vaccination adherence within migrant populations, suggesting the need for multifaceted approaches in public health strategies."

Line 268-269, discussion, "A primary factor influencing lower HPV vaccine uptake is the lack of adequate health information or literacy. This highlights the crucial role of accessible and comprehensive health education in promoting vaccine programs."

Comments on the Quality of English Language

As a non-native English speaker, I have suggested some reformulations to improve the language in the manuscript. However, I recommend considering a review by a native English speaker to further enhance sentence construction, readability, coherence, and accuracy, ensuring a polished final presentation.

Round 2

Reviewer 1 Report

Comments and Suggestions for Authors

The author's response to the third comment was not indicated in the original text, and we did not see any corresponding modifications.

Regarding the eighth comment, the author did not understand my point.

Overall, the article lacks the weighted analysis that should be present in a meta-analysis.
